# Enabling Spatial Data Interoperability through the Use of a Semantic Meta-Model—The Peatland Example from the JRC SEPLA Project

Pavel Milenov [1,†], Aleksandra Sima [2,†], Emanuele Lugato [3], Wim Devos [3] and Philippe Loudjani [3,*]

1 Stalker-KM Ltd., 1463 Sofia, Bulgaria; pavel.milenov@eea.europa.eu
2 Geo-Agri Solutions, 97-300 Piotrków Trybunalski, Poland; aleksandra.sima@eea.europa.eu
3 European Commission, Joint Research Centre (JRC), 21027 Ispra, Italy; emanuele.lugato@ec.europa.eu (E.L.)
* Correspondence: philippe.loudjani@ec.europa.eu
† Current address: European Environment Agency, EEA, 1050 Copenhagen, Denmark.

**Abstract:** Numerous geographic data on peatland exist but definitions vary, and the correspondent classes are often neither harmonized nor interoperable. This hinders the efforts to employ the available national datasets on peatlands and wetlands for policy monitoring and reporting. The existing meta-languages, such as ISO-Land Cover Meta Language (LCML) and EAGLE, offer the possibility to "deconstruct" the relevant nomenclatures in an object-oriented manner, allowing the comparability and interoperable use of related information. The complex nature of peatlands calls for a dedicated and structured vocabulary of keywords and terms, comprising the biotic substrate and the soil. In the SEPLA project, a semantic meta-model has been developed, combining the hierarchical ontology of the LCML with the matrix structure of the EAGLE model. The necessary elements were provided to describe peatland bio-physical characteristics, while representing the definitions in a concise and user-friendly manner (semantic passports). The proposed semantic meta-model is innovative as it enables the documentation of the spatial distribution of peatland characteristics, considering also their temporal dimension, their intrinsic relation with land use, and the soil. It has been successfully implemented for the translation of the national peatland nomenclature into common land categories relevant for reporting under LULUCF regulation, as part of the EU Climate Law.

**Keywords:** meta-language; peatland; satellite data; LULUCF; land mapping; LCML; EAGLE; land characterization

## 1. Introduction

Land Use, Land Use Change and Forestry (LULUCF) is a critical sector for climate change because it offers a strong potential for $CO_2$ removal from the atmosphere.

The European Union (EU) has decided to rely on that sink potential to achieve climate neutrality for the European Union (EU) by 2050. For long, the EU relied largely on forestry and sustainable afforestation, but additional carbon sinks are needed from appropriate agricultural land management. The revised LULUCF regulation (EU Reg. 2018/841) [1] aims to ensure a cohesive policy and effective implementation at national and EU levels. Sustainable agricultural management practices on cropland (such as reduced tillage, occurrence of fallow land, and rotation that includes leguminous crops) and on grassland (ban on ploughing, controlled drainage, or irrigation) preserve carbon already in soils and increase the area that acts as a carbon sink. Next to these, there is an absolute necessity for the protection of healthy peatlands and for remediation actions on degraded ones. The protection, restoration, as well as effective greenhouse gas calculations require up-to-date records that are geographically explicit (e.g., up to individual ecosystems) and that hold information on land use, land use change, land cover, as well as on edaphic conditions. Due

to different reasons, including a lack of agreed conceptualization and a common framework for describing pristine and managed peatlands, no such information currently exists.

The proper definition and mapping of peatlands is a prerequisite for setting-up and performing conservation and restoration efforts, as well as for correctly reporting their carbon fluxes. It requires due consideration on all the various factors that contribute to the resulting complexity [2,3]. Nevertheless, defining, mapping, and monitoring peatlands can be a complex task due to the plethora of peatland definitions [4] and the range of local-specific characteristics that played a role in their genesis [5,6]. Additionally, peatland characteristics, such as vegetation structure, species composition, peat depth and composition, hydrological connectivity, and topography, can vary greatly, which can make their reliable mapping and effective monitoring using Earth Observation (EO) data a challenge [7].

All this hampers the emerging efforts to unlock the potential of the available geographic-explicit national datasets on peatlands and wetlands for policy monitoring and reporting. It also prevents the effective uptake of the relevant spatial data in a machine-readable manner, such as in situ data from training and validation of the EO-based approaches for mapping and monitoring of these areas. A common approach towards the abstracted description of the different types of peatlands in Europe requires both a convention for their formal conceptualization and an agreement on vocabulary. That vocabulary should hold all necessary definitions and axioms that explicate an intended meaning. The existing meta-languages and semantic models, such as ISO-Land Cover Meta Language (LCML) and EAGLE, offer the possibility to "deconstruct", in an object-oriented manner, the relevant classification systems and nomenclatures, allowing comparability and interoperable use of related geographic information. While being proven successful for dealing with land cover nomenclatures [8], it has certain limitations to describe phenomena where the ecological aspects and environment conditions are determining. The complex nature of peatlands and the challenge of their conceptualization [9] in an unambiguous and standardized manner calls for the elaboration of a dedicated and structured vocabulary of keywords and terms, comprising the above-ground biotic substrate, water conditions, and the soil.

In 2021, the Joint Research Centre (JRC) launched the SEPLA (Satellite-based mapping and monitoring of European peatland and wetland for LULUCF and agriculture) project under the work programme with DG CLIMA and technical experts from the national administration and research bodies in 10 EU countries. The main objective of the project was to develop and agree on methods for comprehensive inventories of wetlands and peatlands that support the monitoring of their preservation and restoration. An emphasis was placed on the cross-cutting nature of the available spatial data of peatlands and on the diversity of application domains the data could originate from. A particular challenge of the SEPLA project was to connect the LULUCF accounting to the GAEC (Good Agriculture and Environmental Conditions) standard 2 (on the protection of wetlands and peatlands) in the Common Agricultural Policy (CAP). Where the GAEC 2 focuses on peatlands and wetlands under agricultural management, the LULUCF data inventory should also cover non-agricultural land. The inventory of these areas would require a common perception and definition of what is a wetland or peatland, which has been proven to be challenging [4].

This situation implied that any attempt to propose a common definition for peatlands would result in a compromise between a generic approach suitable for policy reporting purposes and a more specific approach allowing the implementation of the policy objectives at the local level. Such compromise risks rendering the common definition useless in both cases.

Rather than trying to formulate one definition of peatland that is commonly acceptable across EU Member States (MS), the project worked on the elaboration of a common vocabulary of keywords and terms, structured hierarchically by a domain logic, for charting local peatland definitions in an unambiguous and standardized manner via a meta-model. The resulting semantic meta-model had to provide all necessary elements to describe any observable peatland bio-physical characteristic.

The purpose of this meta-model is to enable EU Member States to document their local definitions of peatland and wetland in a standardized manner, and, thus, to be able to perform the following:

(1) Catalogue the thematic data (classes and their mapped instances) in the EU Member States that fell into the scope of SEPLA. These are the wetland and peatland areas under cropland and grassland management, as part of the generic land categories, adopted for reporting under LULUCF;

(2) Identify synergies and differences between each Member State's national geographically explicit dataset based on these definitions;

(3) Compare it with international datasets that can be used to fill gaps in the national data;

(4) Facilitate the selection of the candidate bio-physical characteristics that can be monitored with Earth Observation (EO) technologies.

The main contribution of this paper is a thorough description of the design and implementation of the proposed semantic meta-model enabling the interoperability between different datasets of peatlands. The uniqueness of the semantic meta-model lies in the combination of a structured vocabulary of biophysical properties of all aspects of abov-ground (land cover) and below-ground (soil) substrate. A model extension is also proposed to capture the temporal dimension of the land cover changes and to include the elements of land use characteristics.

After presenting the conceptual framework, the proposed semantic meta-model is described. The reader is next guided through the logic and steps for the creation of a "semantic passport", using, as an example, the "Peat Bogs" class of Copernicus Natura 2000 dataset. The extension of the model towards the spatio-temporal aspect and land use is also explained (Section 3). The implementation of the meta-model in SEPLA in one of the participating countries (Bulgaria) and the results obtained are presented in the following chapter (Section 4.1). The feedback from the implementation of the meta-model in SEPLA and the challenges ahead are given in the Discussion section. At the end of the paper, conclusions and options for further development are provided.

## 2. Conceptual Framework

Peatlands and wetlands are complex phenomena that could be approached from different perspectives: as habitats, as ecosystems, as land cover, as carbon pools, etc. In a spatial data context, peatlands and wetlands are distinct areas characterized by specific biotic, ecological, and edaphic conditions, processes, structure, and functions (life support, water regime regulation, and carbon sequestration). An individual peatland is characterized by a relative uniformity of the physical environment and a close interaction of all its biotic/abiotic components. Peatlands and wetlands can be represented as discrete spatial objects (polygon geometries) derived from a field survey or remote sensing data or depicted or modelled using, as reference, other spatial objects as analytical units, such as grid-cells [10]. The spatial representation and other product specifications depend on the application area, user information needs, and data availability.

The effect of the different peatland definitions (e.g., in [11–13] or national) can be observed in the European peatland map of the Greifswald Mire Centre [14]. The map reflects its sources where national soil data are still very diverse and disparate (e.g., different methods and scales of field survey, different criteria for classifying soils, and different sampling methods and sampling densities), making it difficult to integrate the data meaningfully [15,16].

Therefore, the first step of the work consisted of studying the main characteristics of peatlands. Peatlands and wetlands are two different, albeit overlapping, concepts. There is an agreement that a wetland refers to ecosystems that are water-saturated either permanently (for years or decades) or seasonally (for weeks or months) [17]. They encompass both mineral and organic soils. On the other hand, a peatland refers to ecosystems where the accumulation of organic deposits occurred in water-saturated conditions. Peatlands are complex systems consisting of both biotic and abiotic components, which are very

regional/national-specific. Peatlands are usually considered part of the wetland category, due to the origin and formation that required the persistent presence of water in the past. Peatlands, whether still saturated by water or drained, must have a naturally accumulated layer of peat at the surface.

From a morphological point of view, peatlands represent an intrinsic mix of vegetation and soil elements, involved in a process of material and energy exchange, each with its specific characteristics and properties, subject to change in time. Changes could be part of the natural cycle of the given set of elements or could be due to persistent human intervention. Certain characteristics and properties "manifest" on the surface; thus, they are potentially observable.

Among the bio-physical elements/characteristics that define and describe a peatland, the ones considered fundamental are related to the (1) type of organic deposit, (2) level of water table, (3) type of vegetation cover, and (4) hydrological connectivity (Figure 1).

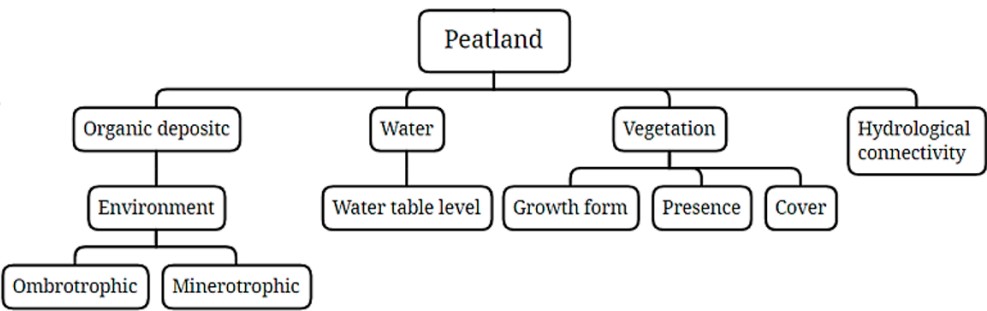

**Figure 1.** Proposed key elements that define a peatland.

The observed diversity and complexity have led to the decision to make use of a semantic meta-model as described hereafter.

## 3. Material and Methods

### 3.1. Design and Demonstration of the Semantic Meta-Model

The design of the semantic meta-model started with an assessment of the various definitions from different expert communities and application areas [11–13] or EU national definitions). This revealed that three aspects appear in all peatland definitions: hydrology, soil, and vegetation. Water plays an obvious central role; the characteristics of the substrate or soil formed under saturated conditions (hydric soils) and the occurrence of vegetation adapted for saturated conditions (hydrophyte vegetation) are key defining aspects for a wetland ecosystem [18].

The semantic meta-model builds on the structured vocabulary of the Land Cover Meta Language (LCML) (ISO 19144-2) [19] and the EAGLE matrix for analytic class decomposition [20]. LCML was found to be particularly suitable for its exhaustive set of classifiers (elements) with tailored properties capable of describing the biotic and abiotic characteristics of the above-ground substrate. LCML also allows the depiction of the vegetation structure through the use of different material strata. Furthermore, it offers a ready-to-use concept for the inclusion of the soil characteristics and the interaction with the water table. LCML logic provides the versatility required to translate, compare, and enable the interaction between different classification systems/nomenclatures. The EAGLE model, on the other hand, provides a way to represent the LCML ontology in tabular form, making it simple and easy to understand for non-experts in domain formalization and semantic modelling. Similarly to LCML, EAGLE offers a tool for the semantic comparison of classes in different classification systems by breaking them down to land cover components, land use aspects, and landscape characteristics. However, due to its purpose to serve mainly EO-based mapping application, it lacks the three-dimensional (strata) representation, provided by the LCML.

The methodological approach used for the design of the SEPLA semantic meta-model for peatlands follows the one used in the land cover eligibility profiles of the Land Parcel Identification System (LPIS) quality assurance framework [21]. The LPIS is the geospatial database supporting the annual farmer CAP subsidy applications that essentially record agricultural activities on agricultural land. An eligibility profile describes the relevant agricultural land for each Member State.

So, the Land Cover Meta Language [19] is used as the core ontology for the design of the semantic "meta-model" containing, in a hierarchical structure, the essential and generally accepted bio-physical characteristics of the cropland, grassland, and wetland, located on peat soils (wet or drained). The ontology contains the morphological topsoil characteristics that could potentially affect the appearance, structure, and other biotic aspects of the land cover. The model is structured by semantic logic. The link between land cover and soil is embedded in the three-dimensional concept of tegon (as a land cover prism) [22] and pedon (as a soil prism) (Figure 2). The introduced connection "tegon–pedon" is composed of the three-dimensional elementary bodies of land cover and soil, respectively. They act as a structural pair in the system of "soil–plant–ground atmosphere" [23]. This innovative concept enables a standardized description of the relationships between the soil and land cover and, in turn, the identification of their biophysical characteristics that can be observed with remote sensing. While preserving the semantic connection between them, it also supports the definition of the "universe of discourse" of land cover (LC) and land use (LU) as two separate thematic areas.

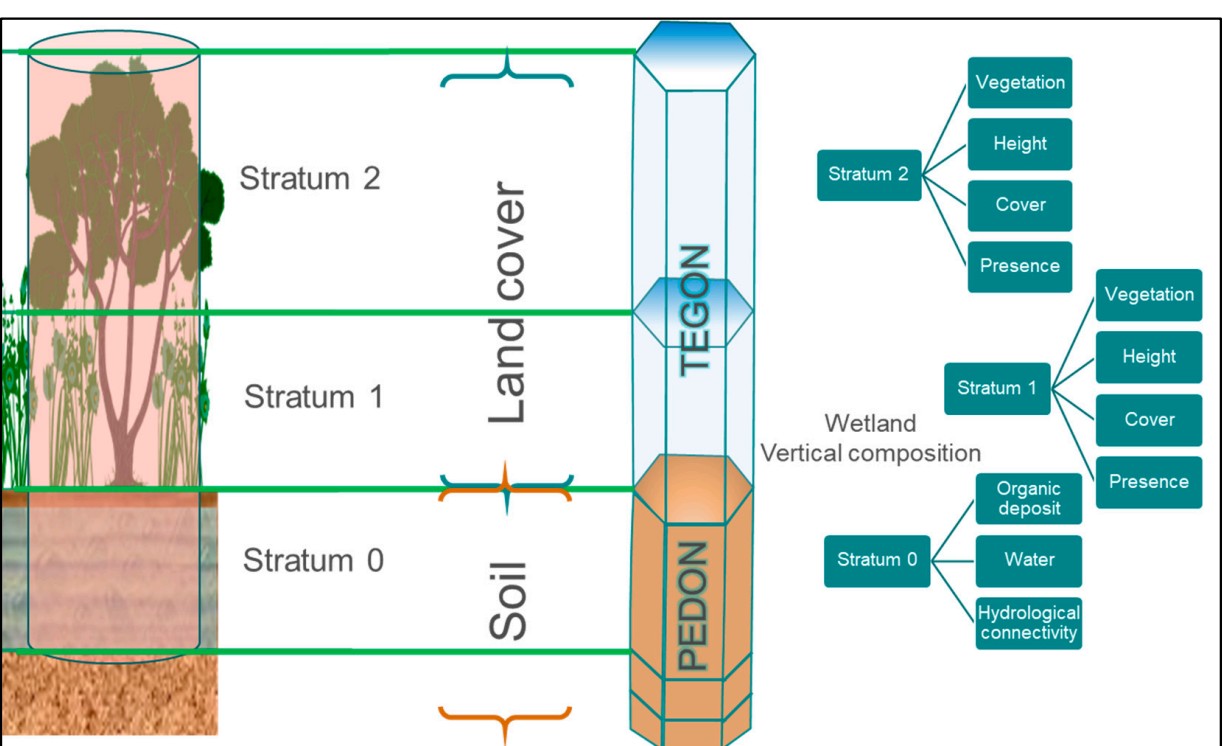

**Figure 2.** Illustration of the developed semantic model concept describing the properties and characteristics of the land cover phenomena and its underlying soil. The biophysical elements are structured through the elementary three-dimensional units of land cover (tegon) and soil (pedon) and their correspondent strata/horizons. The above-ground substrate is described through the vocabulary of LCML, while for the below-ground substrate, a specific vocabulary is designed.

The objective of the semantic model was to document peatlands' biophysical characteristics, regardless of how these are distributed over the site and the cartographic scale of their representation. Consequently, it does reflect neither the spatial heterogeneity of the peatland nor its change over time due to anthropogenic or natural factors. However, it

provides the necessary information to distinguish a real change in the biophysical property due to external factors from a temporal behaviour inherent to its natural life cycle.

The semantic model is realized as a simple table, organized similarly to the EAGLE matrix [20], which comprises the (should be) exhaustive set of LCML-based elements and properties, structured in a hierarchical manner, to characterize the land cover and soil-related aspects of a peatland. As LCML does not cover the soil domain, an extra set of elements and properties to describe the soil characteristics is explicitly introduced in the model. In a typical case of peatland, presented in Figure 2, three vertical layers (strata) can be distinguished. The topsoil stratum (numbered 0) corresponds to the uppermost (water saturated) soil horizon made by organic deposits and having contact with vegetation and the atmosphere. The strata 1 and 2 above the soil (stratum 0) represent the layers of vegetation, typically covered by shrubs and trees, respectively. Each of the strata in the semantic meta-model contains the relevant biophysical elements and properties associated with the biotic or abiotic material present. The semantic meta-model is shown in Figure 3a,b.

The initial proposal for the meta-model was revised with the project partners (four interactions), ensuring correspondence with the logic of the LCML and the EAGLE model applied by the European Environmental Agency, until a stable and coherent version was produced.

The application of the proposed semantic meta-model is intended to be straightforward, and comprises two main stages:

1. Semantic assessment and decomposition of the class definitions, as defined in the related nomenclature. In this step, the textual description of the given class definition is scanned for phrases and key words that relate to elements/characteristics in the meta-model. Once found in the definition, elements/characteristics from the meta-model are flagged and the functional traits between characteristics are highlighted (Figure 4). All classes used to label the objects mapped in a dataset need to be semantically assessed.

2. Assessment of the thematic and quantitative information stored in the dataset, associated with each mapped object (structure and type of mapped objects, input sources, and cartographic scale). In this step, the feature data model is queried for the presence of feature types and attributes, corresponding to the semantic meta-model. Here the importance is to identify the type of quantitative information the given dataset contains. Once found in the feature data model, the characteristics from the meta-model are flagged and the functional traits between characteristics are highlighted.

The meta-model helps to obtain the translated semantic description and to document it systematically as a "semantic passport" of the given class and associated attributive information.

Figure 4 illustrates this passport generation process.

The example in Figure 4 presents the analysis of N2K nomenclature guidelines (version 1.1, 2021) to derive the semantic passport for class 7.1.2 Peat Bogs of the Copernicus N2K (Natura 2000) product.

First, the land-related classes that refer to inland wetlands (class 7.1) on organic soil, including Peat Bogs (class 7.1.2), were identified in the N2K nomenclature. As the N2K nomenclature adheres to the MAES approach (Mapping and Assessment of Ecosystems and their Services, N2K Product User Manual, 2021), these are all classes located in the category "7. Wetland", in the MAES level 1 legend.

That category includes Inland wetlands which are specified at level 2 of the nomenclature. At level 3 of the nomenclature, the class of direct interest is the class 7.1.2 "Peat bogs", with its two sub-classes (level 4) "7.1.2.1 Exploited peat bog" and "7.1.2.2 Unexploited peat bog" (Figure 5).

(a)

(b)

**Figure 3.** (**a**) Semantic meta-model produced in the frame of the SEPLA project to describe different peatland types with details for the strata 1 and 2 representing lower and higher vegetation, respectively. (**b**) Semantic meta-model produced in the frame of the SEPLA project to describe different peatland types with details for the topsoil stratum (stratum 0). The additional table (lower right) shows the key aspects of the local environment playing a role in the peatland genesis.

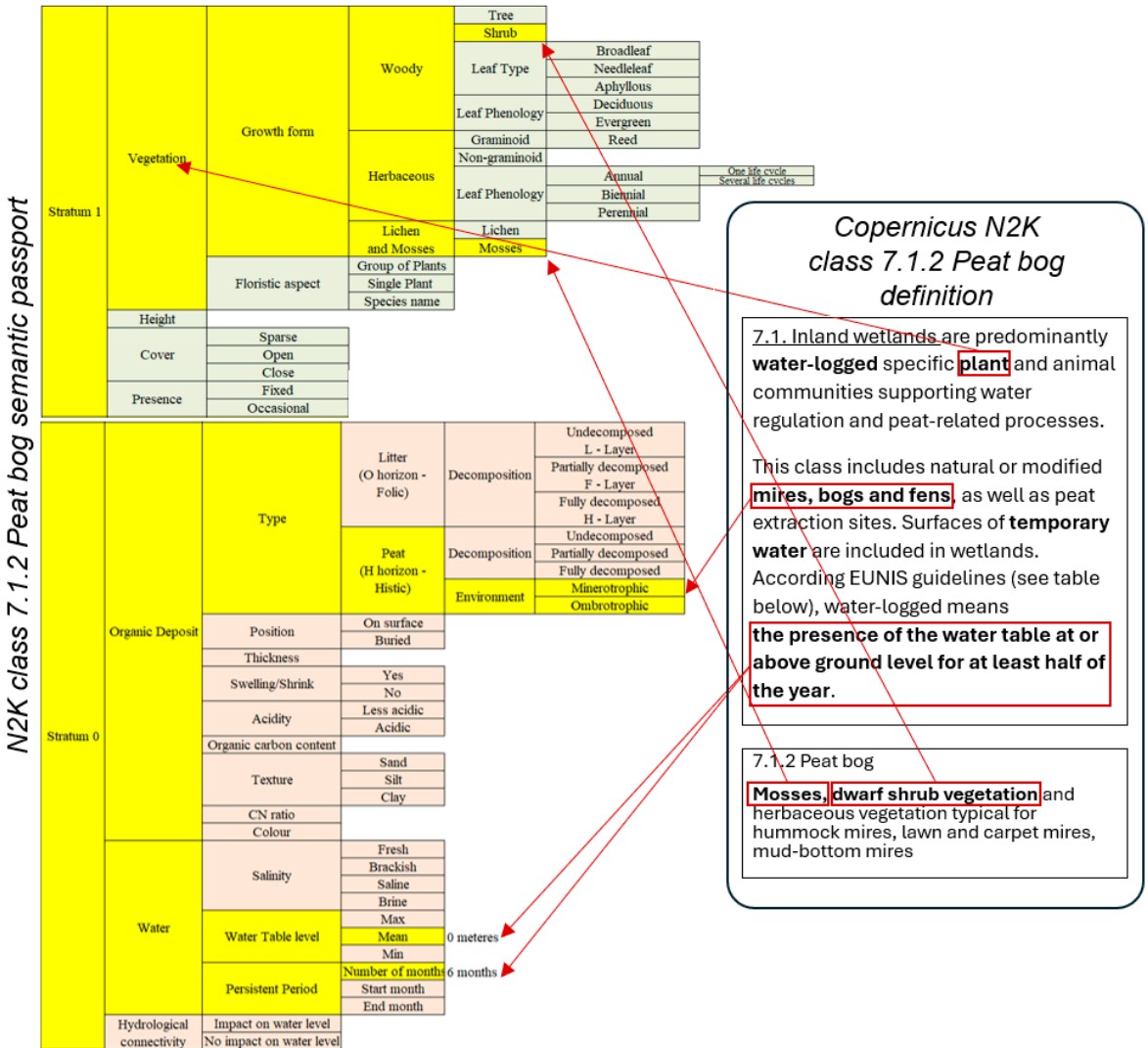

**Figure 4.** Illustration of the decomposition of the 7.1.2. Peat bog class (a subclass of 7.1 Inland wetland class) of the Copernicus N2K nomenclature to obtain a corresponding "semantic passport".

The elaborated semantic meta-model is applied to identify the characteristics from the class documentation corresponding to the generic peatland semantic meta-model. The process starts from the upmost hierarchical category and goes down to lower levels to collect extra information on the type of land under each sub-category.

In the N2K nomenclature, the "7. Wetland" class definition states that it concerns the "inland freshwater/saline wetlands" only. The key words and expressions in that definition that can be linked to the ones in the 'wetland semantic template' are identified (and evidenced in italic bold in the text below).

Inland wetlands are predominantly "water-logged" specific "plant" and animal communities supporting water regulation and peat-related processes. This class includes natural or modified "mires, bogs and fens", as well as "peat" extraction sites (MAES). The surfaces of "temporary water" are included in wetlands. According to EUNIS guidelines, water-logged means the "presence of the water table at or above ground level for at least half of the year" [24].

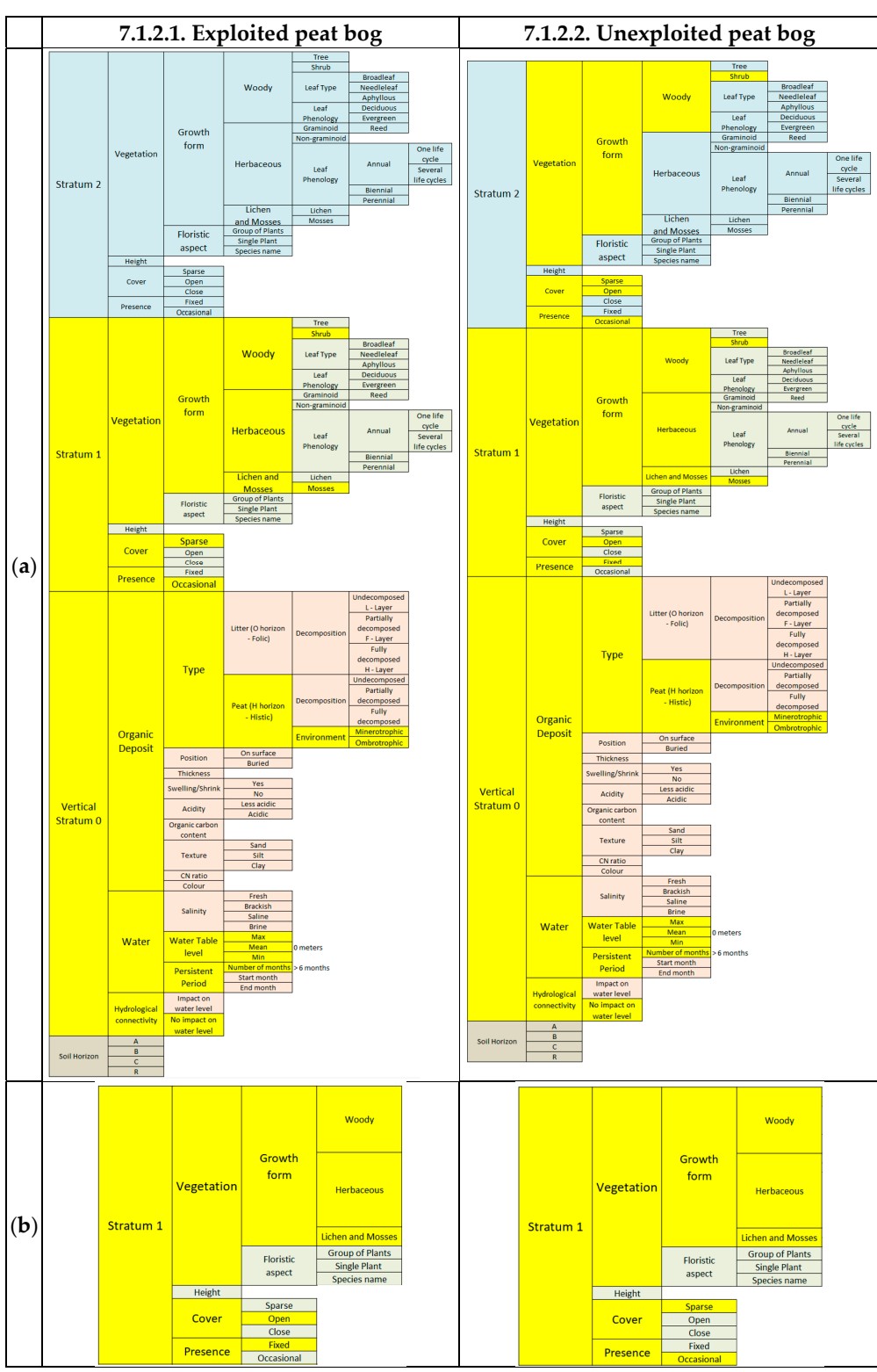

**Figure 5.** *Cont.*

| (c) | Landform | Mountain |
| | | Hill |
| | | Plateau |
| | | Plain |
| | Topography | Altitude |
| | | Slope |
| | Geography | Inland |
| | | Coastal |
| | Climate | Boreal |
| | | Cold temperate dry |
| | | Cold temperate wet |
| | | Warm temperate dry |
| | | Warm temperate moist |
| | | Mediterranean |

| Landform | Mountain |
| | Hill |
| | Plateau |
| | Plain |
| Topography | Altitude |
| | Slope |
| Geography | Inland |
| | Coastal |
| Climate | Boreal |
| | Cold temperate dry |
| | Cold temperate wet |
| | Warm temperate dry |
| | Warm temperate moist |
| | Mediterranean |

**Figure 5.** Semantic passports of the classes "Exploited peat bog" and "Unexploited peat bog" of the Copernicus N2K nomenclature. Section (**a**) shows the entire set of land cover elements of the passports. This overview highlights the main difference between the two classes—the lack of Stratum 2 for "Exploited peat bog". Section (**b**) shows the differences between the two classes for Stratum 1, sparse cover and occasional presence of low vegetation for "Exploited peat bog", in contrast with open cover and fixed presence for "Unexploited peat bog". Section (**c**) shows the main differences with respect to the "contextual" aspects.

In the semantic template, the keyword "peat" is used to highlight (see yellow highlights in Figure 4) and fill the cells for [Stratum 0], [Organic Deposit] -> [Type] -> [Peat (H horizon—Histic)] -> [Environment] -> [Minerotrophic], and [Ombrotrophic]. Because no information on the levels of decomposition is available, the cells in the [Organic Deposit] functional trait are not highlighted.

In [Stratum 0], the keywords/phrases "water-logged", "water table", "at or above ground level" and "at least half of the year" are utilized to highlight the cells [Water] -> [Water Table level] -> [Mean] and provide the value "0 m" next to the [Mean]. Also, the cells [Water] -> [Persistent Period] -> [Number of months] cells are selected and the value ">6 months" is provided in the empty cell next to the [Number of months].

The definition contains the word "plant", further defined as the vegetation that could be found on peatlands: "Mosses, dwarf shrub vegetation and herbaceous vegetation typical for hummock mires, lawn and carpet mires, mud-bottom mires".

In the semantic meta-model, these keywords allow the ability to highlight and fill cells in [Stratum 1], [Vegetation] -> [Growth form] -> [Herbaceous]. In the absence of information about the leaf phenology or type of herbaceous plant, cells in the [Vegetation] functional trait are left blank. Then, [Vegetation] -> [Growth form] -> [Lichen and Mosses] -> [Mosses] is highlighted together with [Vegetation] -> [Growth form] -> [Woody] -> [Shrub] (see Figure 4).

The semantic analysis has been further extended using information from the two peatland sub-classes "7.1.2.1. Exploited peat bog" and "7.1.2.2. Unexploited peat bog" and the obtained semantic passports are illustrated in Figure 5.

## 3.2. Extension towards the Temporal Dimension and Land Use

The physical appearance of any biotic land cover at a given time will depend on its geographic location; the physiognomy, structure, and phenology of vegetation; and the impact of anthropogenic or natural events. For wetlands, observable changes in state (and area) might result from natural evolution (phenology cycle), a modification of its natural environment (climate change), or human intervention (e.g., drainage).

The key to success for the impact assessments of land policies (which all target human interventions) will depend on the ability to distinguish natural evolution from changes induced by human activity. Another innovative part of the work is that it has extended the capabilities of semantic assessment to account for the impact of land use and natural disturbances on peatland's behaviour in time. Describing human activity is the scope of the Land Use Meta Language (LUML), which is still under discussion and development (future ISO 19144-3). However, it is equally important to document the very relationship between land use (LU) and land cover (LC), which is the aim of the ongoing ISO initiative to translate and register the existing classification systems (future ISO 19144-4). Elements

from the two developing ISO initiatives and the SEPLA semantic meta-model have been used to elaborate a structured template for documenting the association and interaction between farming activities and bio-physical aspects of the agricultural land cover. It served the thorough documentation of Checks by Monitoring (CbM) implementations [25] in the Common Agricultural Policy by introducing the notion of "activity" to describe an anthropogenic intervention and "event" to describe a natural impact.

The logic of the structured CbM template was further used in the SEPLA project to formalize the interactions between (1) the bio-physical characteristics of the peatland (land cover) under agricultural management and (2) activities (land use) at the level of the individual biotic component and its vertical strata (Figure 6).

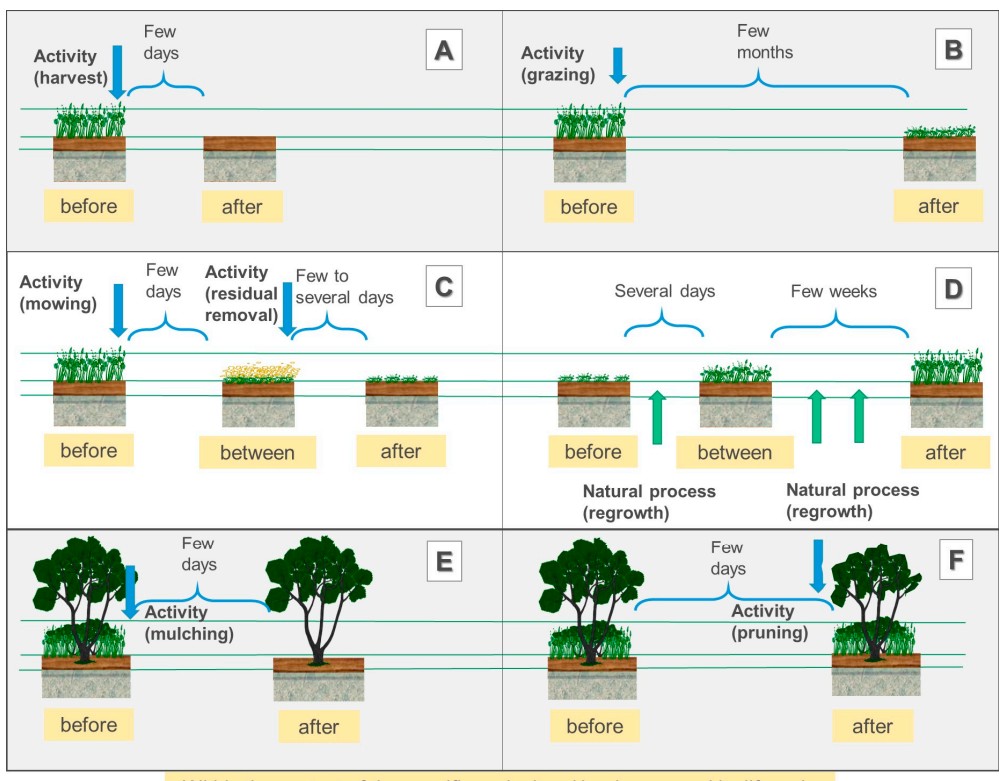

**Figure 6.** Examples of manifested changes in state following (**A**) harvest of arable crop, (**B**) grazing, (**C**) mowing, (**D**) grass regrowth, (**E**) mulching, and (**F**) pruning. The blue arrows indicate the occurrence of an activity; the green arrows indicate the occurrence of a natural process. Source: JRC.

The GeoSpatial Application (GSA), a formal annual submission of a farmer applying for EU CAP subsidy, serves as the information source for the anticipated sequence of activities during the agronomic season (scenario). The structured CbM template then gathers information on the nature and the timings of the bio-physical "manifestations" (LC) of these activities to allow for their wide spatial (automated) capturing and mapping by a given EO sensor. The integration of the structured CbM template in the peatland monitoring process will facilitate the tracking of the peatland status and the expected impact of any restoration efforts applied.

### 3.3. Implementation Setup of the Semantic Assessment in SEPLA

The objectives of the semantic assessment in the SEPLA project were to (1) help EU Member States to make an inventory of their national wetland and peatland datasets; (2) identify those classes depicting wetland and peatland areas under cropland and grassland management and those that are in its natural state (pristine); and (3) select and apply the spatial operations on the datasets associated with these classes to produce the ge-

ographic explicit data in line with LULUCF reporting requirement. Experts from four EU Member States (Bulgaria, Denmark, Ireland, and Latvia) have performed a thorough assessment of the nomenclatures of their national datasets (land cover, protected habitats, and digital soil maps) on peatlands and wetlands. Both the SEPLA project team and EU country experts then assessed the class "passports" in relation to the following:

- Correspondence to LULUCF information needs/requirements;
- Semantic gaps and overlaps (including data availability check);
- Role of the class/dataset in the spatial data integration flow for the generation of the candidate peatland data, in line with the generic categories relevant for the LU-LUCF reporting;
- Identification of those classes, indicating peatlands at risk or degraded.

The semantic passports were created for each of the relevant classes from the assessed nomenclatures. Then, the semantic passports were checked for the presence of the following characteristics/classifiers: (1) organic matter/soil organic carbon; (2) persistent occurrence of water; (3) vegetation strata; and (4) information on land use. A check of the semantic passports for consistency and obvious errors was conducted in parallel. For those "passports", holding explicit information of the temporal aspects of the vegetation behaviours and wetness (seasonality of water, phenology, and farming practices), additional verification on up-to-date Sentinel-2 (Level 1-C), and historic Landsat TM imagery were conducted.

While the analysis was based on the information provided in the semantic passports, it also consulted the correspondent data models and product specification to understand and document the process of class instantiation. For each class, its potential role and value to derive the candidates for wetland and peatland in a LULUCF-relevant context (grassland on drained peatland, cropland on drained peatland, pristine peatland, pristine wetland) was evaluated.

## 4. Results and Discussion

### 4.1. Results

The whole process and the analysis outcome are illustrated here in the detailed example of a case study in Bulgaria. The country has a scarce presence of peatlands, but a high share of natural protected areas associated with wetlands (Figure 7). Two sites have been selected, DRAG in the Sofia plain measuring 25 km × 25 km, and RILA in the mountain range, south of the city, measuring 50 km × 50 km. They have been selected due to the different types of peatlands occurring in the sites, their distinct topography, hydrology, and climate. Prior documented studies over the area [26] were an added asset. In the DRAG site, the focus was on the "Dragoman Marsh", being in a state of advanced restoration after agricultural use for the last 30 years. In the RILA site, the focus was on the peatlands located in the sub-alpine and alpine belt of the Rila Mountain used for extensive grazing.

Two national datasets were chosen for the study: the N2000 dataset and the Digital Soil Map (Figure 8). The semantic assessment identified five Habitat Directive (HD) Annex I classes, potentially associated with inland peatland and four soil mapping units with known areas correlated with histosols. Class descriptions of the habitats were taken from the national N2000 mapping manual [27]. The definition correlated with the updated 2017 EUNIS habitat classification (https://eunis.eea.europa.eu/index.jsp accessed on 1 April 2021). Class descriptions of the soil units were derived from reference literature [28,29]. Semantic passports were created for each of the classes based on these class definitions alone. The main working assumption was that the candidate peatland areas for LULUCF reporting could be derived through a spatial union of the designated peat-related habitats under N2000 and the areas related to histosols.

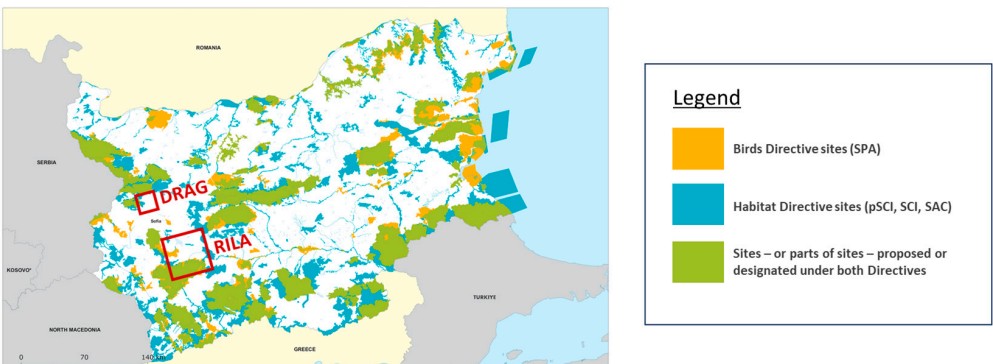

**Figure 7.** Distribution of Natura 2000 sites in Bulgaria (reference year 2020). The locations of two test sites are marked in red. Source: European Environment Agency. Map compiled based on data from DG ENV, Eurogeographics, and DG ESTAT.

| Habitat code | Habitat description |
|---|---|
| 7140 | Transition mires and quaking bogs |
| 7230 | Alkaline fens |
| 6410 | Molinia meadows on calcareous, peaty or clayey-silt-laden soils (Molinion caeruleae) |
| 3160 | Natural dystrophic lakes and ponds |
| 91D0 | Bog woodland |

(a)

| National code | Description |
|---|---|
| БТС | Histosols (terric) |
| БТ | Histosols (fibric) |
| ХТС | Histosols (terric), salty |
| ХТФ | Histosols (fibric), salty |

Note: The soil nomenclature and soil characteristics are correlated to those of FAO.

(b)

**Figure 8.** (**a**) Selected Annex I habitats related to peat; (**b**) Selected SMUs correlated with histosols.

The results of the semantic assessment revealed that classes 7140, 7230, and 91D0 are clearly related to peatlands and were directly considered candidates for "pristine peatland". Further investigation was necessary for class 6410, representing meadows on or next to peat, and for class 3160, representing water bodies with peat at their bottom. There were also additional Habitat Directive Annex I habitat classes identified, such as 3150 (natural eutrophic lakes) that could potentially be peatland candidates.

The two soil map units correlated with histosols were considered candidates for "drained peatland", while those correlating with salty histosols correlated with the IPCC sub-category "pristine peatland". The spatial assessment of the soil map unit features from the soil dataset revealed that histosols do not always correspond to the identified N2000 peatland types. Based on plant communities, hydrological conditions, and location, they could be associated with other non-peat HD Annex I habitat classes (3150, 6150, 1150, and 1530).

The spatial discrepancy between wetlands on peat and histosols was due to the conceptual difference in the nomenclatures. Wetland classes describe mainly ecosystems and plant communities, while soil classes describe soil genesis and morphology and provide information on the associated vegetation (Figure 9).

The spatial data (class instantiation) associated with the selected wetland classes and histosols were ingested into a stepwise spatial data integration flow for the generation of the candidate peatland data, in line with generic categories relevant for the LULUCF reporting (Figure 10). It comprised four main steps organized into two main process traits:

(1) Generation of the spatial extent of the inland pristine peatlands:

    a. Step 1: Derive the "union" of the spatial extent of the N2000 wetland classes related to peat with the spatial extent of histosols;

    b. Step 2: Derive the "intersection" between the spatial extent of any other N2000 wetland classes and the spatial extent of the salty histosols;

    c. Combine the results of Step 1 and Step 2.

(2) Generation of the spatial extent of the inland drained peatlands:

a. Step 3: "Subtract" from the spatial extent of the salty histosols those areas that intersect with the spatial extent of the N2000 wetland classes related to peat;
b. Step 4: "Subtract" from the spatial extent of the salty histosols those areas that intersect with the spatial extent of any other N2000 wetland classes;
c. Combine the results of Step 3 and Step 4.

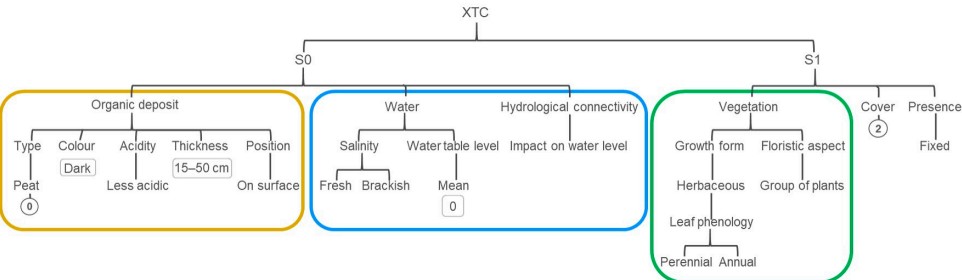

**Figure 9.** Graphical illustration (spider diagram) of the semantic passport for soil class XTC with the soil, water, and vegetation characteristics highlighted in different colours. The spider diagram is an alternative graphical representation of the semantic passport of a class.

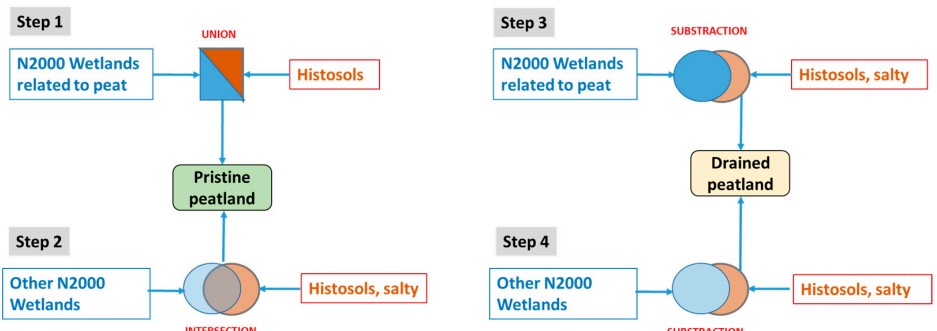

**Figure 10.** Graphical illustration of the spatial data integration flow for the generation of the candidate peatland data in line with IPCC categories.

As no spatial features associated with salty histosols were present in the section of the soil map covering the Bulgarian test sites, only step 1 was performed.

The resulting candidate peatlands associated with the category of "pristine peatland" (marked in brown) are presented in Figure 11, on top of a land cover map, derived from the LPIS (year 2021). Only 1% of agricultural parcels in 2021 intersect with the peatland areas.

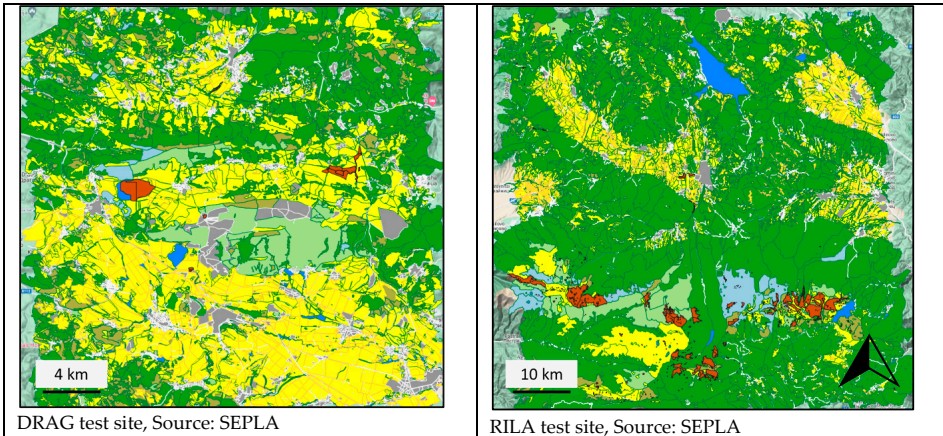

**Figure 11.** Resulted spatial extent of the pristine peatlands (in brown) for the two test sites in Bulgaria. The forests are marked in dark green; sparse natural vegetation in light green; cropland, grassland, and permanent crops in yellow; water bodies in blue; and other land cover classes in grey.

*4.2. Discussion*

The proposed method has been tested by several experts from four EU Member States using the datasets (maps, legends) of their territory. The successful translation and exchange of information from the different local situations by the semantic meta-model demonstrated its usefulness as a practical solution for effective class documentation and knowledge exchange and made the datasets interoperable. The test employing multiple national datasets also allowed the project to identify and fill-in some description gaps (e.g., how to describe peat buried by upper soil horizon) and to elucidate semantic overlaps.

The semantic meta-model and the generated semantic passports allow the identification of those classes from independent local (national) datasets (land cover, wetland habitats, and soil maps), which are relevant as peatland areas in the context of the LULUCF and CAP-GAEC 2 legislations. Once a class is identified, its spatial distribution is ready to be included in any assessment of peatland areas. In addition, the standardized description of the class provides a better understanding of the nature of the land cover and features occurring in those peatland-related areas.

The semantic meta-model facilitates comparisons and distinctions between some subclasses, as completed for the classes "Unexploited peat bog" and "Exploited peat bog" of N2K Copernicus product discussed in Material and Methods and illustrated in Figure 5. The first is broad and includes practically all types of bogs and fens of the different climatic zones in Europe and is covered by a range of vegetation types. The second class refers to a specific situation of peatlands under a particular type of management. They are further characterised by a lack of persistence of herbaceous vegetation and by the absence of woody plants. Information on the occurrence and persistence of the water level would allow for the separation of both "peat bog" classes from other wet areas such as the wet (mesic) grasslands. The assessment process delivers associated ancillary information such as the soil characteristics (soil organic content, texture, thickness, etc.).

The presented semantic assessment formalizes the connection between human activity and the affected set of observable biophysical characteristics. It further allows for the association of the latter with appropriate Earth Observation data and methods. This is a key step towards designing a unified methodology that would facilitate the broad assessment and the mapping of state and change in state of peatlands under the restoration efforts by any EU rules (as presented in the Results section).

In addition, the experiment demonstrated that the SEPLA semantic meta-model can be further extended with land use components to further subdivide a peatland/wetland by the type of management and land custodian. It is completed through the combination of corresponding spatial data, such as the LPIS and the farmer's annual GeoSpatial Application, or the digital maps of habitats under protection, and will result in spatial/physical subdivisions of the peatland extent into "single units of management". Their abstraction is comparable with the concept of Feature of Interest (FOI) that was introduced in the frame of the CAP Checks by Monitoring. A FOI is the physical surface of the Earth where a specific agricultural practice is planned and performed [30]. In farming business terms, this surface will correspond to a particular cropped plot, fenced grassland, or an orchard. In the SEPLA context, the FOI concept was reused to represent managed or natural peatlands and wetlands surfaces (Figure 12).

Figure 12 depicts various types of FOIs, relevant for different policies and different monitoring purposes. The red outline delimits the entire Natura 2000 area. It is composed of pristine wetland (marked in orange) and two types of managed grasslands, as declared in the GeoSpatial Application; some pastures are used for grazing (in green) and some meadows (in violet) are mowed. The manifested land cover changes in both types of grasslands will differ due to different agricultural practices applied (see Figure 6B,C). This resulting spatial division and semantic description is now expected to allow for the use of EO methods to comprehensively monitor the state of the entire Natura 2000 area.

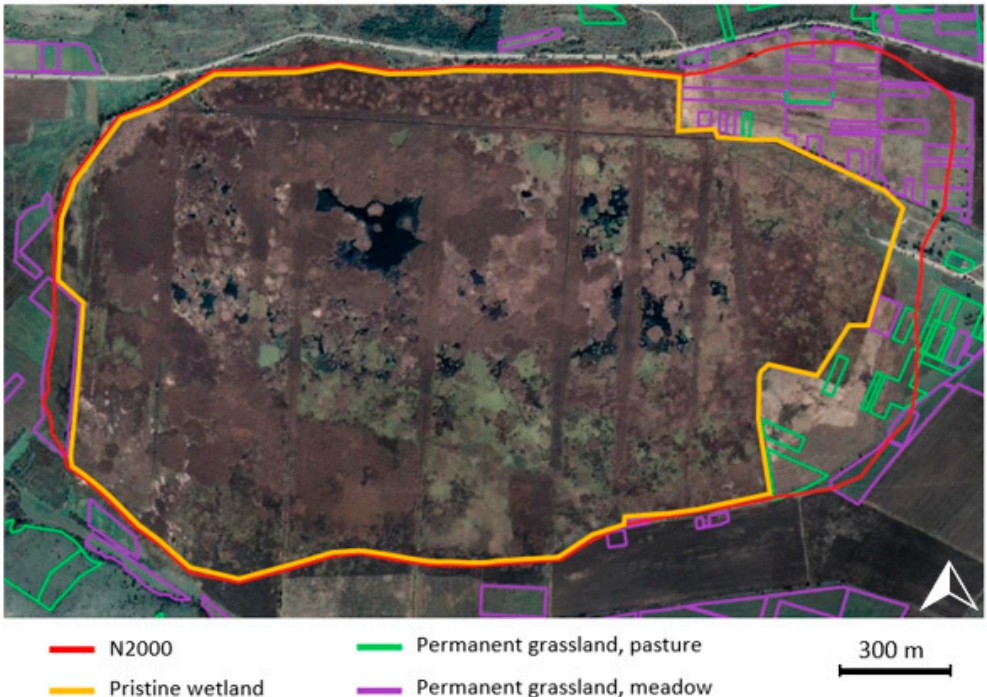

**Figure 12.** The Dragoman Marsh is the biggest natural karst wetland in Bulgaria, included in N2000, and located on soils that are predominantly histosols. Most of its area is a close-to-pristine state after its restoration, but it also comprises limited areas of permanent grasslands still used for agriculture. Outlines of the permanent grasslands come from the LPIS, and the land use is declared in the farmer's GeoSpatial Application. The N2000 outline comes from the dedicated mapping of the marsh as a specific habitat, under Annex I of the Habitat Directive. Image: Google, 2023.

## 5. Conclusions

The importance of the clear identification and quantification of peatlands for LULUCF reporting made it a prominent use case to tackle some ubiquitous challenges for compiling a comprehensive inventory of specific areas. It requires addressing the local specificities of the landscape and socio-economic context, the variety of non-harmonized and non-interoperable data, and the lack of integration among the different "expert communities" dealing with the considered land domain. The semantic meta-model, developed within the SEPLA project, provided the necessary elements to describe any observable peatland characteristic and allowed the translation of the national peatland nomenclature into common land categories relevant for policy reporting, such as the LULUCF.

The SEPLA semantic meta-model was successfully applied by EU MS's experts to perform the "semantic mapping" of their existing datasets, produce "semantic passports", and compile their datasets for LULUCF reporting. The provided feedback evidenced that the model is rigorous, as the determining factors or diagnostic criteria based on ISO-LCML and EAGLE are objective, logical, and generally accepted. The model is easy to understand and practical to use since it does not require a high level of technical expertise.

The innovative aspect of the proposed semantic meta-model is a joint description of above-ground (land cover) and below-ground (soil) substrates though the use of the tegon–pedon concepts, allowing for an integrated criteria-based assessment of peatland ecosystems. It aligns land cover concepts with those of the more familiar soil ontology. Another novel element is the explicit introduction of a temporal dimension of land cover's life cycle and its connection with land use.

The standardized semantic "passport" produced for each described peatland helped the identification of synergies and differences between each Member State's national geographically explicit dataset based on these definitions. It further allows the comparison with

international datasets and facilitates the selection of candidate bio-physical characteristics that can be monitored with Earth Observation (EO) technologies.

Future studies will focus on improving the capability of the semantic meta-model to describe the spatial relationship between the individual elements (vegetation, soil, and water) in intrinsically mixed classes, using the "horizontal pattern" mechanism of LCML.

The SEPLA project was implemented in parallel with the revision of the LCML [19], which provided an opportunity for cross-fertilization and project contribution to the ISO developments. The SEPLA project findings contributed to the improved capability of LCML to describe the 3D aspect of a complex land cover class with multiple vertical strata. The project produced some real-world examples of class instantiations and LC–LU interactions, which were included in the standard.

Using FOI with Earth Observation data proved essential to gather and document information on spatial heterogeneity and the spatial propagation of land change. The integration of the semantic meta-model with the FOI concept and the structured CbM template will be tested beyond the agriculture and wetland domains (e.g., forests and natural grasslands) as part of the development of future ISO 19144-3 and ISO 19144-4.

**Author Contributions:** Methodology, P.M., A.S., E.L. and W.D.; Data curation, P.L.; Writing—review & editing, P.M., A.S., E.L., W.D. and P.L. All authors have read and agreed to the published version of the manuscript.

**Funding:** This research received an internal European Commission funding through an Administrative Arrangement between the DG CLIMA and DG JRC.

**Data Availability Statement:** The original contributions presented in the study are included in the article, further inquiries can be directed to the corresponding author.

**Acknowledgments:** The authors wish to acknowledge the valuable contributions from the technical experts in Denmark, Ireland, and Latvia for sharing their methodological experience and, especially, from Bulgaria for sharing the illustration used in this document.

**Conflicts of Interest:** The authors declare that the research was conducted in the absence of any commercial or financial relationships that could be construed as a potential conflict of interest.

**Disclaimer:** The views expressed in this article are solely those of the authors and its content does not necessarily represent the views or position of the European Environment Agency.

## Acronyms

| | |
|---|---|
| CAP | Common Agriculture Policy |
| CbM | Checks by Monitoring |
| EAGLE | Action Group on Land monitoring in Europe of the European environment information and observation network |
| EC | European Commission |
| EEA | European Environmental Agency |
| EO | Earth Observation |
| EU | European Union |
| FOI | Feature of Interest |
| GAEC | Good Agriculture and Environmental Condition |
| GSA | Geo Spatial Application |
| HD | Habitat Directive |
| IACS | Integrated Administration and Control System |
| IPCC | Intergovernmental Panel on Climate Change |
| ISO | International Organization for Standardization |
| JRC | Joint Research Centre |
| LCML | Land Cover Meta Language |
| LC | Land Cover |

| LPIS | Land Parcel Identification System |
|------|-----------------------------------|
| LU | Land Use |
| LUML | Land Use Meta Model |
| LULUCF | Land Use, Land Use Change and Forestry |
| MAES | Mapping and Assessment of Ecosystems and their Services |
| MS | Member State |
| N2000 | Natura 2000 sites network |
| SEPLA | Satellite based mapping and monitoring of European peatland and wetland for LULUCF and agriculture |
| SMU | Soil Map Unit |

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
