# Peer review of "Enabling Spatial Data Interoperability through the Use of a Semantic Meta-Model—The Peatland Example from the JRC SEPLA Project"

_land, doi:10.3390/land13040473_

Round 1

Reviewer 1 Report

Comments and Suggestions for Authors

This research paper considers a situation where geographic data on peatland exists, but descriptions vary, and the classes involved are often incompatible and unconnected. In order to solve this problem, a common vocabulary of words and terms that describe peatlands in a clear and standard way has been developed. Below is a list of revisions needed.

1. The article highlights the diversity and nonstandardization of peatland definitions. However, more details are needed on how effective common vocabularies of words and terms are, which is the proposed solution to overcome this problem. The criteria on which these terms were selected and what standard was established require further clarity.

2. What is the SEPLA? and What is EAGLE model? Should be explained with details.

3. More information is needed about how comprehensive and flexible the semantic meta-model outlined in the article is. It is not specified which features this model focuses on and which data types it supports.

4. Box concept: The authors created something complex in terms of general meaning. It contains structures that need to be displayed separately, such as figures, text, and tables, which makes reading difficult.

5. Figure 4 and Figure 5 are not understandable.

6. It is stated that experts have successfully tested the proposed semantic meta-model. Still, these testing processes should include more details on what criteria are considered a success and what difficulties arise.

7. The contribution of ISO 19144-2 to this work is emphasized, but concrete examples or details are not given as to what kind of improvements these contributions specifically lead to.

8. The research's similarity report contains a 24% match. This may be acceptable by journal standards, but even if it is, I think it is necessary to make the manuscript original. If an expression that must be presented without changes, such as a law or an aphorism, is not used, a source should not be quoted in an almost identical sentence. It is unacceptable to take verbatim sentences, especially from the source "https://wikis.ec.europa.eu/display/GUIDANCEANDTOOLSFORCAP" (18% match is seen in the similarity report).

9. Some of the sources used in the text have been used only once. No discussion took place. Additionally, there were many self-citations for one of the article's authors. There are a total of 12 articles in the reference list, and self-citation is for three articles (25% of all citations are to this author).

10. The reference list is numerically insufficient, as some of the sources are old and have lost direct relevance to the subject's context. I recommend that more detailed references be created by scanning the literature.

Comments on the Quality of English Language

Must improve the sentences.

Author Response

Dear reviewer, 

We would like to thank you for your time and effort in reviewing our paper. We are pleased to note that these comments and suggestions were all valuable and very helpful for revising and improving our manuscript. We have studied them carefully and tried to address all the weaknesses you mentioned.  

We also mention that since we made substantial changes to the first version, the revised version is provided without track changes to ease the reading.

Best regards 

The authors

Reviewer 2 Report

Comments and Suggestions for Authors

The authors proposed a semantic meta-model to unambiguously describe the variety of peatland definitions. The authors used the EAGLE model in the context of the JRC SEPLA project. The topic is relevant and crucial to discussing the semantic meaning of LULC classes, especially in areas susceptible to biodiversity loss and damage to ecosystem services. However, the piece needs a deeper organization of its sections and a deeper review of updated literature.

A few specific comments:
1) Introduction: I suggest to include some updated references here. There are no references cited along the 3-paragraphs of the Introduction. I suggest including a deeper literature review on peatland definitions in current mapping and modeling initiatives across the EU.
2) The author must define the SEPLA project for a broader audience of the Land Journal.
3) The authors must present the primary goal of the piece better. Please ensure that the piece is about wetlands in general or peatlands in the EU. In my opinion, the methodology and results presented do not achieve the primary goal. The piece's goal is presented again, in different ways, in line 128. Please make sure the primary goal of the manuscript is meaningful.
4) Please check Figure 1. Peatland is the only type of wetland, and it does not make sense in terms of ontological specifications.
5) There is also a lack of an updated literature review in the Material and Methods section.
6) Box 1 is mixed with the text, and it is unclear what it means or how it connects with Figure 3. In general, the entire Material and Methods section must be reorganized.
7) The Results and discussion are weak and focused on the methodology. The broader discussion section must be incorporated.
8) The ISO issue is challenging to understand, which made the manuscript much more descriptional and less scientific.

Author Response

Dear reviewers, 

We would like to thank you for your time and effort in reviewing our paper. We are pleased to note that these comments and suggestions were all valuable and very helpful for revising and improving our manuscript. We have studied them carefully and tried to address all the weaknesses you mentioned.  

We also want to mention that since we made substantial changes to the first version, we submit the second version without the track changes to ease the reading.

Best regards

The authors

Round 2

Reviewer 1 Report

Comments and Suggestions for Authors

Thanks for your efforts and best wishes for your manuscript.

Reviewer 2 Report

Comments and Suggestions for Authors

All my comments were adequately accommodated.